# Assessment of the Usefulness of Cefapirin and Cefalonium Disks for Susceptibility Testing of *Staphylococcus aureus* Isolates from Bovine Mastitis

**DOI:** 10.3390/antibiotics9040197

**Published:** 2020-04-21

**Authors:** Kazuki Harada, Shieri Irie, Mamoru Ohnishi, Yasushi Kataoka

**Affiliations:** 1Department of Veterinary Internal Medicine, Tottori University, Tottori 680-8553, Japan; 2Laboratory of Veterinary Microbiology, Nippon Veterinary and Life Science University, Tokyo 180-8602, Japan; shieri2011@gmail.com (S.I.); ykataoka@nvlu.ac.jp (Y.K.); 3Ohnishi Laboratory of Veterinary Microbiology, Hokkaido 086-1106, Japan; oawcn953@ybb.ne.jp

**Keywords:** bovine mastitis, antimicrobial disk, staphylococci, cefapirin, cefalonium

## Abstract

Cefapirin (CEP) and cefalonium (CNM) are first-generation cephalosporins widely used to treat bovine mastitis caused by Gram-positive bacteria including staphylococci. However, disks for susceptibility testing of those drugs in causative bacteria are not available. This study evaluated the efficacy of 10 µg and 30 µg pilot disks of CEP (CEP10 and CEP30) and CNM (CNM10 and CNM30) against 130 *Staphylococcus*
*aureus* isolates from bovine mastitis. Scattergrams of minimum inhibitory concentrations (MICs) and zone diameters (ZDs) illustrated significant correlations between the MICs and ZDs of CEP10 (*r* = −0.912), CEP30 (*r* = −0.933), CNM10 (*r* = −0.847), and CNM30 (*r* = −0.807). The analysis by Normalized Resistance Interpretation indicated that the epidemiolocal cut-off value (ECV) of MIC for both cefapirin and cefalonium is ≤ 0.5 µg/mL, and the ECV of ZD for CEP10, CEP30, CNM10, and CNM30 were ≥ 22 mm, ≥ 25 mm, ≥ 22 mm, and ≥ 29 mm, respectively. We believe that both 10 μg and 30 μg CEP and CNM susceptibility disks will be helpful for guiding the appropriate use of these antibiotics for bovine mastitis. Further studies toward the establishment of clinical breakpoint of CEP and CNM would be needed for their routine use.

## 1. Introduction

Bovine mastitis is a common disease of dairy cows worldwide, causing decreased milk production, increased of veterinary care cost, and increased culling [1,2]. Most mastitis cases result from intramammary infections caused by *Staphylococcus aureus*. Coagulase-negative staphylococci (CNS) may also cause mastitis, but CNS are less pathogenic than *S. aureus* [3,4,5]. Antimicrobials are the primary treatment of bovine mastitis, but overuse and misuse of antibiotics have been associated with the emergence of bacterial resistance and the entrance of resistant bacteria into the food chain [6,7]. Against this background, it is strongly recommended that the choice of the antimicrobial drugs used to bovine mastitis should be based on the antimicrobial susceptibility of the causative staphylococcal strain [8].

Cefapirin and cefalonium are first-generation cephalosporins that are widely used to treat bovine mastitis caused by Gram-positive bacteria including staphylococci [9,10,11,12]. Resistance to β-lactams including penicillin and cephalosporins by staphylococci isolated from bovine mastitis has been increasing, and the prevalence of methicillin-resistant *S. aureus* (MRSA) in bovine mastitis is of concern in veterinary medicine and public health worldwide [8,13,14]. Consequently, test tools that can quickly and easily detect cefapirin and cefalonium resistant bovine mastitis infections are essential for appropriate antibiotic use. This pilot study evaluated the efficacies of cefapirin, cefalonium, and cefazolin susceptibility disks for *S. aureus* isolates from bovine mastitis based on the hypothesis that the cefapirin and cefalonium disks have high utility for their susceptibility testing, as well as the disk of cefazolin as a standard of cephalosporin drug.

## 2. Results and Discussion

This study evaluated the usefulness of 10 µg and 30 µg containing disks of cefapirin (CEP10 and CEP30, respectively) and cefalonium (CNM10 and CNM30, respectively), together with cefazolin disk (CEZ30), for susceptibility testing of *S. aureus* isolates from bovine mastitis. Table 1 shows the MIC distribution for the tested three drugs. The MICs of cefazolin and cefalonium were trimodally distributed, whereas that of cefapirin was bimodally distributed. The analysis by Normalized Resistance Interpretation (NRI) [15,16] indicated that the epidemiological cut-off values (ECVs) were lower for cefapirin (≤0.5 µg/mL) and cefalonium (≤0.5 µg/mL) than for cefazolin (≤2 µg/mL) (Table 1). Likewise, the mode of MIC was lower for both cefapirin (0.25 µg/mL) and cefalonium (0.125 µg/mL), than for cefazolin (0.5 µg/mL). Thus, cefapirin and cefalonium have consistently been shown to have higher bactericidal activity for *S. aureus* isolates from bovine mastitis than cefazolin. 

The ranges of the zone diameters (ZDs) of the five disks by MICs of the corresponding drug are shown in Table 2, and scattergrams of ZDs and MICs of these drugs are shown in Figure 1. The minimum and maximum ZDs were 6–44 mm for CEP10, 10–51 mm for CEP30, 7–44 mm for CNM10, and 14–51 mm for CNM30. The regression-line equations were y = 68.1 − 1.98x (*r* = −0.912) for CEP10, y = 79.3 − 2.02x (*r* = −0.933) for CEP30, y = 28.7 − 0.85x (*r* = −0.847) for CNM10, and y = 32.6 − 0.86x (*r* = −0.807) for CNM30. The slope and intercept values of the 10 µg and 30 µg disks of each antibiotic, were similar and all of the correlation coefficients for the MIC and ZD values were significant, as well as cefazolin (*r* = −0.900, *p* < 0.001). As the results, the zone diameters of both 10 µg and 30 µg cefapirin and cefalonium disks have high correlation with their MICs, as well as that of cefazolin disk.

We further estimated ECVs of ZD for all disks by using NRI method, and then obtained following values: ≥27 mm for CEZ30, ≥22 mm for CEP10, ≥25 mm for CEP30, ≥22 mm for CNM10, and ≥29 mm for CNM30 (Figure 1). When ECVs of MIC and ZD were applied to our collection, several strains fell into different categories between MIC and ZD. One strain (0.8%) had higher MIC and longer ZD than each ECV of CEZ30, CEP10, CEP30, or CNM10, whereas six strains (4.6%) had lower MIC and shorter ZD than each ECV of CNM30. Although these discrepancy rates are relatively low, further studies would be desired to verify the validity of ECVs established in this study.

## 3. Materials and Methods

A total of 130 *S. aureus* isolates, consisted of 54 MRSA strains and 76 methicillin-susceptible *S. aureus* (MSSA) strains, collected from dairy cows with bovine mastitis in Japan between 2010 and 2011 were included in the analysis. 

Sampling and bacterial isolation from milk samples was carried out according to the protocol of National Mastitis Council [17]. Briefly, milk samples were obtained from mammary papilla after sterilized with alcohol. A loopful of milk samples was streaked on Pourmedia Sheep Blood Agar (Eiken Chemical Co., Ltd., Tokyo, Japan), and then incubated aerobically at 36 °C for 48 h.

The bacterial strains were identified by colony morphology, hemolysis pattern and Gram staining as previously described [18]. Gram-positive cocci were tested for catalase and oxidase and *S. aureus* isolates were identified by species-specific polymerase chain reaction, as previously described [19]. Methicillin resistance was confirmed with an MRSA-LA latex agglutination assay (Denka Seiken, Tokyo, Japan) that detects penicillin-binding protein 2’ [20]. All bacterial strains were stored at −80 °C in 10% skim milk.

Agar disk diffusion testing was performed following CLSI guidelines [21]. The following five antibiotics disks were used: The 30 μg cefazolin disk (KB Disk Eiken Cefazolin, Eiken Chemical Co., Ltd., Tochigi, Japan), and the four pilot disks prepared by Eiken Chemical Co., Ltd., i.e., 10 μg and 30 μg cefalonium disks (CNM10 and CNM30, respectively) and 10 μg and 30 μg cefapirin disks (CEP10 and CEP30, respectively). The isolates were suspended in sterile saline at a concentration corresponding to a McFarland 0.5 turbidity standard. The bacterial suspensions were inoculated onto Mueller–Hinton agar with a cotton swab. After placing the disks on the agar, the plates were incubated at 35 °C in ambient air for 18 h. The diameter of the inhibition zone measured with calipers. *S. aureus* ATCC 25923 was used as a quality-control strain. 

Agar dilution testing was performed following CLSI guidelines [21]. The three cephalosporin antibiotics, cefazolin, cefapirin, and cephalonium, were tested in serial 2-fold dilutions ranging from 0.031 to 2048 μg/mL in Mueller–Hinton agar. The plates were inoculated with a 1 mm pin multipoint applicator that delivering approximately 1.0 × 10^4^ colony-forming units per spot and then were incubated at 35 °C in ambient air for 20 h. The MICs were determined as the lowest concentration of drugs that completely inhibited colony formation. *S. aureus* ATCC 29213 was used as a quality-control strain.

We estimated ECVs based on distributions of MICs or ZDs for the tested drugs by using the NRI method [15,16], which was used with permission from the patent holder, Bioscand AB, TÄBY, Sweden (European patent No 1383913, US Patent No. 7,465, 559). In this method, the automatic and manual excel programmes (2019 version) were made available through courtesy P. Smith, W. Finnegan, and G. Kronvall. The scattergrams with ZDs on the *X*-axis and MICs on the *Y*-axis for each drug were constructed using all of 130 *S. aureus* isolates. Based on the scattergrams, the significance of correlations between the ZDs and MICs of each drug was confirmed by regression analysis and Pearson correlation coefficients was calculated. Regression lines were calculated, excluding off-scale ZDs that were ≤6 mm in diameter as described previously [22]. *p*-values <0.05 were considered significant. 

## 4. Conclusions

Cefapirin and cefalonium susceptibility testing of *S. aureus* isolates from bovine mastitis resulted in scattergrams showing that the ZDs of both 10 μg and 30 μg disks of the two drugs were significantly correlated with the MICs of each tested antibiotic. The correlation analysis indicated that both 10 μg and 30 μg cefapirin and cefalonium disks were effective for estimating drug susceptibility. We believe that these disks of cefapirin and cefalonium are helpful for appropriate use of these antibiotics for treating bovine mastitis. In addition, we proposed that the ECV of MIC for both cefapirin and cefalonium is ≤0.5 µg/mL, and the ECV of ZD for CEP10, CEP30, CNM10, and CNM30 were ≥22 mm, ≥25 mm, ≥22 mm, and ≥29 mm, respectively, based on each distribution. However, such ECVs would be weighted towards microbial population distributions rather than towards clinical outcomes [23]. Unfortunately, mastitis-specific clinical breakpoints have not been established for staphylococcal isolates except for ceftiofur and pirlimycin [21,24]. Future studies of the clinical breakpoints of cefapirin and cefalonium are needed for the effective use of these antibiotic disks.

## Figures and Tables

**Figure 1 antibiotics-09-00197-f001:**
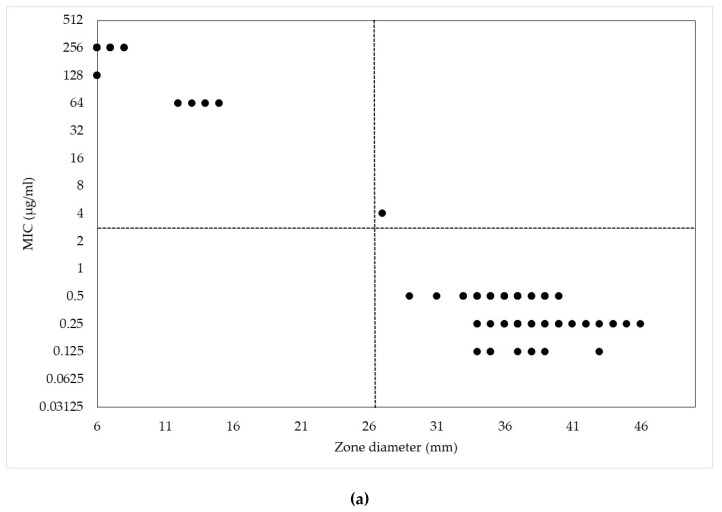
Scattergrams of the minimum inhibitory concentrations (MICs) and zone diameters (ZDs) of the tested cephalosporin drugs for 130 *S. aureus* isolates from bovine mastitis. (**a**) Cefazolin MICs versus CEZ30 ZDs; (**b**) Cefapirin MICs versus CEP10 ZDs; (**c**) Cefapirin MICs versus CEP30 ZDs; (**d**) Cefalonium MICs versus CNM10 ZDs; (**e**) Cefalonium MICs versus CNM30 ZDs. The dotted lines mean the ECVs established in this study. Each marker may include several strains.

**Table 1 antibiotics-09-00197-t001:** The MICs of cefazolin, cefapirin, and cefalonium for 130 *Staphylococcus aureus* isolates from bovine mastitis.

Antimicrobials	MIC (µg/mL)													
	0.031	0.063	0.125	0.25	0.5	1	2	4	8	16	32	64	128	256
Cefazolin	−	−	8	28 (1)	43 (2)	−	−	1 (1)	−	−	−	7 (7)	3 (3)	40 (40)
Cefapirin	−	1	34 (2)	40 (1)	4	1 (1)			1 (1)	6 (6)	4 (4)	39 (39)	−	−
Cefalonium	3	12 (1)	57 (1)	7 (1)		1 (1)		7 (7)	1 (1)	25 (25)	17 (17)	−	−	−

The vertical lines mean the epidemiolocal cut-off values (ECVs) estimated by using the Normalized Resistance Interpretation (NRI) method [15,16]. The numbers in parenthesis mean the number of MRSA strains.

**Table 2 antibiotics-09-00197-t002:** The range of ZDs of CEZ30, CEP10, CEP30, CNM10, and CNM30 disks by MICs of the corresponding drug for 130 *S. aureus* isolates from bovine mastitis.

MIC (μg/mL)	Range of Zone Diameters (mm)				
	CEZ30	CEP10	CEP30	CNM10	CNM30
256	6–8	−	−	−	−
128	6	−	−	−	−
64	12–15	6–9	10–14	−	−
32	−	8–9	12–14	7–14	14–17
16	−	12–13	18–21	8–14	14–19
8	−	18	20	14	20
4	27	−	−	18–21	24–26
2	−	−	−	−	−
1	−	23	28	26	26
0.5	29–40	26–34	33–37	−	−
0.25	34–46	28–37	32–42	26–29	28–33
0.125	34–43	29–44	33–51	26–38	27–44
0.063	−	39	45	28–44	32–51
0.031	−	−	−	36–43	39–48

Cefazolin MIC was applied for CEZ30 disk, cefapirin MIC for CEP10 and CEP30 disks, and cefalonium MIC for CNM10 and CNM30 disks. The horizontal lines mean the ECVs of MIC estimated by using the NRI method [15,16].

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
