# Peer review of "Assessment of the Usefulness of Cefapirin and Cefalonium Disks for Susceptibility Testing of *Staphylococcus aureus* Isolates from Bovine Mastitis"

_antibiotics, 2020, doi:10.3390/antibiotics9040197_

Round 1

Reviewer 1 Report

The authors are to be praised for undertaking an important task which will provide valuable data to a crucial research field of antimicrobial resistance. The study evaluated the efficacy of antibiotic disks of the two most widely used antimicrobials cefapirin and cefalonium. Cefapirin 30 μg (CEP30) and cefalonium 30 μg (CNM30) disks had significantly large zone diameters (ZDs) than CEP10 and CNM10. The study also evidenced a significant correlation between the MICs and ZDs. The results indicated that the cefapirin and cefalonium susceptibility disks will be helpful for guiding the appropriate use of these antibiotics for bovine mastitis.

The manuscript is well written and has well described methodology.

There is a minor query from my side about the following point-

The authors concluded in line 172: “the 10-ug cefapirin disk and the 30-ug cefalonium disk might be preferable to the other contents.”

However, the statements in the line-94-95 and line-99-100 seem to be contradictory to the concluding remark. In line 94-95: “Our study results indicate that 30 ug is also appropriate for cefapirin but not for cefalonium” and in line 99-100: “Based on those findings, we concluded that 10-ug was an effective content for use in a cefalonium sensitivity disk for staphylococcal isolates.”

Minor typos:

A hyphen (-) is appeared between a number and the unit (e.g. line 49: “10-μg and 30-μg”) and sometime no hyphen (e.g. Line 94: “30 μg”). I suggest following a homogenous style throughout the text.

Author Response

Dear Reviewer 1:

We are grateful to Reviewer 1 for the critical comments and useful suggestions that have helped us to improve our paper. As indicated in the responses that follow, we have taken all these comments and suggestions into account in the revised version of our paper as much as possible.

Comment:

The authors concluded in line 172: “the 10-ug cefapirin disk and the 30-ug cefalonium disk might be preferable to the other contents.”

However, the statements in the line-94-95 and line-99-100 seem to be contradictory to the concluding remark. In line 94-95: “Our study results indicate that 30 ug is also appropriate for cefapirin but not for cefalonium” and in line 99-100: “Based on those findings, we concluded that 10-ug was an effective content for use in a cefalonium sensitivity disk for staphylococcal isolates.”

Response:

Thank you very much for your critical comment, and then we are terribly sorry for these misdescriptions. As you indicated, these results were likely to be confused. We finally confirmed that the 30 ug cefapirin disk and the 10 ug cefalonium might be preferable to the other contents based on the present results of correlation analysis. Based on the conception, the following sentences was revised.

(Line 22-24)

Statistical analysis indicated slightly better performance of CEP30 and CNM10 than the other content disk of each drug, although both content disks had high correlation coefficients.

(Line 75-77)

The results imply that although both the 10 µg and 30 µg cefapirin and cefalonium disks were effective for estimating drug susceptibility, the 30 µg cefapirin disk and the 10 µg cefalonium disk might be preferable to the other contents.

(Line 151-154)

The correlation analysis indicated that although both the 10 μg and 30 μg cefapirin and cefalonium disks were effective for estimating drug susceptibility, the 30 μg cefapirin disk and the 10 μg cefalonium disk might be preferable to the other contents.

Comment:

A hyphen (-) is appeared between a number and the unit (e.g. line 49: “10-μg and 30-μg”) and sometime no hyphen (e.g. Line 94: “30 μg”). I suggest following a homogenous style throughout the text.

Response:

Thank you very much for invaluable comment. In response to your comment, all hyphens in the text were deleted. Please confirm the revised manuscript.

Finally, we also revised our manuscript according to Reviewer 2’s comments.

We greatly appreciate your help concerning improvement to this paper. We hope that the revised manuscript is now acceptable for publication.

Yours very sincerely,

Kazuki Harada, DVM, Ph. D.

Reviewer 2 Report

Manuscript ID: antibiotics-769149

General comments:

Very interesting and applicable subject. Authors could describe the problem of mastitis in a wider scope and emphasise the importance and applicable value of the study.

Detailed comments:

  • Give the research hypothesis
  • In Abstract use CEP and CNM abbreviations – line 22
  • Fig 1 a and b - they are the same …?
  • Authors should improve the way of presenting the results which are in Figure 1; they are little informative and illegible – what authors wanted to present?
  • What is the correlation between ZD – MIC - and staphylococcal species? Is some?  Give table or graph.
  • Authors should give more detailed description of Materials and Methods - it should be divided into paragraphs – in that form it is confusing.
  • How the strains were isolated? How the samples were taken? Why the strains were not genetically identified (16S rRNA)?
  • What substance was tested as a positive control for agar disc diffusion and for agar dilution testing?

Author Response

Dear Reviewer 2:

We are grateful to Reviewer 2 for the critical comments and useful suggestions that have helped us to improve our paper. As indicated in the responses that follow, we have taken all these comments and suggestions into account in the revised version of our paper as much as possible.

Comment:

Give the research hypothesis.

Response:

Thank you for the invaluable comment. In reply to your comment, the following sentence in the Introduction part was revised.

(Line 45-48)

This pilot study evaluated the efficacies of cefapirin, cefalonium, and cefazolin susceptibility disks for staphylococcal isolates from bovine mastitis based on the hypothesis that the disks of cefapirin and cefalonium are comparable in capability of the susceptibility assessment to the cefazolin disk.

Comment:

In Abstract use CEP and CNM abbreviations – line 22

Response:

In reply to your comment, the descriptions of ‘cefapirin’ and ‘cefalonium’ in Abstract were replaced with CEP and CNM, respectively. Please confirmed the revised manuscript.

Comment:

Fig 1 a and b - they are the same …?

Authors should improve the way of presenting the results which are in Figure 1; they are little informative and illegible – what authors wanted to present?

Response:

Thank you very much for your critical comment, and then we are terribly sorry for mistakes in Figure 1. As you indicated, Fig 1 (a) was mistakenly the same as Fig1 (b), and originally different figure. On the other hand, Fig 1 (b) was correct. In Figure 1, we would have liked to present the relationship between zone diameter and MICs in each drug. As you indicated, the new Table 2 was constructed to make these data more concise and clear. Alternatively, Figure 1 was converted to the supplementary materials (i.e. Figure 1S). In addition, the following sentences were revised. Please confirm the revised manuscript.

(Line 64-65)

The ranges of the zone diameters (ZDs) of the five disks by MICs of the corresponding drug are shown in Table 2, and scattergrams of ZDs and MICs of these drugs are shown in Figure 1S.

(Line 70-72)

The slope and intercept values of the 10 µg and 30 µg disks of each antibiotic, were similar and all of the correlation coefficients for the MIC and ZD values were significant (P < 0.001, Table 3).

(Line 107-108)

The isolates included 54 MRSA strains, 76 methicillin-susceptible S. aureus (MSSA) strains, and 43 CNS strains (Table 4).

Comment:

What is the correlation between ZD – MIC - and staphylococcal species? Is some?  Give table or graph.

Response:

Thank you very much for your comment. In this study, the correlation between ZD and MIC were analyzed based on the scattergrams (i.e. new Figure 1S). All of 173 staphylococcal isolates were used in the correlation analysis. The distribution of staphylococcal species in used isolates was shown in new Table 4. To clarify this point, the following sentence was revised. We would appreciate it if you would clarify the additional Table and/or Figure we should give you.

(Line 140-143)

The scattergrams with ZDs on the X-axis and MICs on the Y-axis for each drug were constructed using all of 173 staphylococcal isolates (Figure 1S). Based on the scattergrams, the significance of correlations between the ZDs and MICs of each drug and the between the ZDs of cefapirin or cefalonium and cefazolin were confirmed by regression analysis.

Comment

Authors should give more detailed description of Materials and Methods - it should be divided into paragraphs – in that form it is confusing.

How the strains were isolated? How the samples were taken? Why the strains were not genetically identified (16S rRNA)?

Response:

Thank you for your indication. In reply to your comment, the first paragraph in Materials and Methods was divided into three paragraphs including the following new paragraph. In this study, method of sampling and bacterial isolation was carried out according to National Mastitis Council. To clarify this point, the following sentences and reference were added.

(Line 109-112)

Sampling and bacterial isolation from milk samples was carried out according to the protocol of National Mastitis Council [22]. Briefly, milk samples were obtained from mammary papilla after sterilized with alcohol. A loopful of milk samples was streaked on Pourmedia Sheep Blood Agar (Eiken Chemical Co., Ltd., Tokyo, Japan), and then incubated aerobically at 36°C for 48 h.

(Line 222-223)

  1. National Mastitis Council. Laboratory handbook on bovine mastitis. Revised edition.; National Mastitis Council: Madison, WI., 1999.

As you indicated, genetic identification is more reliable than phenotypic identification for CNS from bovine mastitis (Zadoks et al., 2009; Park et al., 2011). Unfortunately, however, bacterial identification in this study was carried out in workplace for clinical examination, where any genetic testing devices equipped. We would appreciate your understanding on this circumstance. But we agree with your comment. Thus, we decided to remove identification results of CNS from Table 4, and revised the corresponding sentence according to the suggestion by Zadoks and Watts (2009) that reporting as coagulase-negative Staphylococcus species may be more appropriate than reporting of more detailed but potentially inaccurate results when phenotypic identification was applied.

(Line 116-118)

CNS species were identified by an API Staph System (BioMérieux Japan Ltd., Tokyo, Japan), but were collectively dealt with as CNS in this study, because of the potential inaccurate identification of this kit [25,26].

In addition, the following references were added.

(Line 229-233)

  1. Zadoks, R.N.; Watts, J.L. Species identification of coagulase-negative staphylococci: genotyping is superior to phenotyping. Vet. Microbiol. 2009, 134, 20-28.
  2. Park, J.Y.; Fox, L.K.; Seo, K.S.; McGuire, M.A.; Park, Y.H.; Rurangirwa, F.R., Sischo, W.M., Bohach, G.A. Comparison of phenotypic and genotypic methods for the species identification of coagulase-negative staphylococcal isolates from bovine intramammary infections. Vet. Microbiol. 2011, 147, 142-148.

Comment:

What substance was tested as a positive control for agar disc diffusion and for agar dilution testing?

Response:

In this study, S. aureus ATCC 25923 strain and ATCC 29213 strain were used as quality control strain, according to CLSI guidelines. To clarify this point, the following sentences were described.

(Line 131-132)

The diameter of the inhibition zone measured with calipers. S. aureus ATCC 25923 was used as a quality-control strain.

(Line 138-139)

  1. aureus ATCC 29213 was used as a quality-control strain.

Finally, we also revised our manuscript according to Reviewer 1’s comments.

We greatly appreciate your help concerning improvement to this paper. We hope that the revised manuscript is now acceptable for publication.

Yours very sincerely,

Kazuki Harada, DVM, Ph. D.

Round 2

Reviewer 2 Report

I am satisfied with all the answers.

Author Response

Dear Academic Editor:

We are grateful to the Academic Editor for the critical comments and useful suggestions that have helped us to improve our paper. As indicated in the responses that follow, we have taken all these comments and suggestions into account in the revised version of our paper as much as possible.

Comment:

Lines 25-26 and 157-159. Can the authors use their CMI data for a proposal of tentative microbiological cut-off values for Staph. aureus against these antibiotics? Maybe this could be mentioned in line 54 after the description of the CMI distributions.

Response:

Thank you very much for your critical comment. We fully agree with your comment. In fact, the setting of epidemiological cut-off value (ECV) is important for susceptibility testing of the antimicrobial drug, although ECV is distinguished from clinical breakpoint used for treatment decisions. Thus, we decided to consider ECV for cefapirin and cefalonium. Firstly, as you indicated, we dealt with data only about S. aureus in revised manuscript because it is recommended that ECV should be set based on one species rather than several species. To carry out this revision, we removed all data about CNS throughout the manuscript including title to focus only S. aureus. Secondly, we estimated ECVs for not only MIC but also zone diameters by using the Normalized Resistance Interpretation method. Thirdly, we added the relevant sentences to elucidate and discuss the data on ECVs. The above-mentioned revisions were as follows.

(Line 2-4)

Assessment of the usefulness of cefapirin and cefalonium disks for susceptibility testing of Staphylococcus aureus isolates from bovine mastitis

(Line 16-18)

This study evaluated the efficacy of 10 µg and 30 µg pilot disks of CEP (CEP10 and CEP30) and CNM (CNM10 and CNM30) against 130 Staphylococcus aureus isolates from bovine mastitis.

(Line 18-21)

Scattergrams of minimum inhibitory concentrations (MICs) and zone diameters (ZDs) illustrated significant correlations between the MICs and ZDs of CEP10 (r = −0.912), CEP30 (r = −0.933), CNM10 (r = −0.847), and CNM30 (r = −0.807).

(Line 21-24)

The analysis by Normalized Resistance Interpretation indicated that the epidemiolocal cut-off value (ECV) of MIC for both cefapirin and cefalonium is ≤ 0.5 µg/ml, and the ECV of ZD for CEP10, CEP30, CNM10, and CNM30 were ≥ 22 mm, ≥ 25 mm, ≥ 22 mm, and ≥ 29 mm, respectively.

(Line 55-58)

The analysis by Normalized Resistance Interpretation (NRI) [15,16] indicated that the epidemiological cut-off values (ECVs) were lower for cefapirin (≤ 0.5 µg/ml) and cefalonium (≤ 0.5 µg/ml) than for cefazolin (≤ 2 µg/ml) (Table 1).

(Line 63-64)

Table 1. The MICs of cefazolin, cefapirin, and cefalonium for 130 S. aureus isolates from bovine mastitis.

(In Table 1, number of isolates were revised to describe only S. aureus strains)

(Line 71-73)

The regression-line equations were y = 68.1 - 1.98x (r = -0.912) for CEP10, y = 79.3 - 2.02x (r = -0.933) for CEP30, y = 28.7 - 0.85x (r = -0.847) for CNM10, and y = 32.6 - 0.86x (r = -0.807) for CNM30.

(Line 78-85)

We further estimated ECVs of ZD for all disks by using NRI method, and then obtained following values: ≥ 27 mm for CEZ30, ≥ 22 mm for CEP10, ≥ 25 mm for CEP30, ≥ 22 mm for CNM10, and ≥ 29 mm for CNM30 (Figure 1). When ECVs of MIC and ZD were applied to our collection, several strains fell into different categories between MIC and ZD. One strain (0.8%) had higher MIC and longer ZD than each ECV of CEZ30, CEP10, CEP30, or CNM10, whereas six strains (4.6%) had lower MIC and shorter ZD than each ECV of CNM30. Although these discrepancy rates are relatively low, further studies would be desired to verify the validity of ECVs established in this study.

(Line 86-87)

Table 2. The range of ZDs of CEZ30, CEP10, CEP30, CNM10, and CNM30 disks by MICs of the corresponding drug for 130 S. aureus isolates from bovine mastitis.

(In Table 2, the data were revised to focus only on S. aureus strains)

(Line 89-90)

The horizontal lines mean the ECVs of MIC estimated by using the NRI method [15,16].

(Line 113-115)

A total of 130 S. aureus isolates, consisted of 54 MRSA strains and 76 methicillin-susceptible S. aureus (MSSA) strains, collected from dairy cows with bovine mastitis in Japan between 2010 and 2011 were included in the analysis.

(Line 142-146)

We estimated ECVs based on distributions of MICs or ZDs for the tested drugs by using the NRI method [15,16], which was used with permission from the patent holder, Bioscand AB, TÄBY, Sweden (European patent No 1383913, US Patent No. 7,465, 559). In this method, the automatic and manual excel programmes (2019 version) were made available through courtesy P. Smith, W. Finnegan, and G. Kronvall.

(Line 158-162)

In addition, we proposed that the ECV of MIC for both cefapirin and cefalonium is ≤ 0.5 µg/ml, and the ECV of ZD for CEP10, CEP30, CNM10, and CNM30 were ≥ 22 mm, ≥ 25 mm, ≥ 22 mm, and ≥ 29 mm, respectively, based on each distribution. However, such ECVs would be weighted towards microbial population distributions rather than towards clinical outcomes [23].

(Line 208-211)

  1. Joneberg, J.; Rylander, M.; Galas, M.F.; Carlos, C.; Kronvall, G. Analysis of parameters and validation of method for normalized interpretation of antimicrobial resistance. Int. J. Antimicrob. Agents 2003, 21, 525-535.
  2. Kronvall, G.; Giske, C.G.; Kahlmeter, G. Setting interpretive breakpoints for antimicrobial susceptibility testing using disk diffusion. Int. J. Antimicrob. Agents 2011, 38, 281-290.

(Line 227-228)

  1. Silley, P. Susceptibility testing methods, resistance and breakpoints: what do these terms really mean? Rev. sci. Off. int. Epiz. 2012, 31, 33-41.

Comment:

Line 43. I do not agree with the classification of “on-farm” for antimicrobial susceptibility test, since this test must be performed at laboratory level, not on-farm.
Response:

We agree with your comment, and then the sentence was revised as follows.

(Line 44-46)

Consequently, test tools that can quickly and easily detect cefapirin and cefalonium resistant bovine mastitis infections are essential for appropriate antibiotic use.

Comment:

Lines 47-48. Although the need for a hypothesis was suggested by one reviewer, this statement is not clear.

Response:

In response to your comment, the sentence was revised as follows.

(Line 46-49)

This pilot study evaluated the efficacies of cefapirin, cefalonium, and cefazolin susceptibility disks for S. aureus isolates from bovine mastitis based on the hypothesis that the cefapirin and cefalonium disks have high utility for their susceptibility testing, as well as the disk of cefazolin as a standard of cephalosporin drug.

Comment:
Line 50. Although English is not my mother tongue, “availability” does not seem to be the more appropriated word for this sentence.
Response:

In response to your comment, the sentence was revised as follows.

(Line 51-53)

This study evaluated the usefulness of 10 µg and 30 µg containing disks of cefapirin (CEP10 and CEP30, respectively) and cefalonium (CNM10 and CNM30, respectively), together with cefazolin disk (CEZ30), for susceptibility testing of S. aureus isolates from bovine mastitis.

Comment:

Line 54. I appreciate very much the use of the number of modes for describing the data distribution.
Response:

In response to your comment, the sentence was revised as follows.

(Line 58-59)

Likewise, the mode of MIC was lower for both cefapirin (0.25 µg/ml) and cefalonium (0.125 µg/ml), than for cefazolin (0.5 µg/ml).

Comment:

Lines 55-58. The comparison of MIC50 and MIC90 values from different antibiotics has a very scarce interest.

Response:

In response to your comment, the data on MIC50 and MIC90 was removed from text and Table 1. Please check the revised manuscript.

Comment:
Line 65. “Scattergrams” or “Scatter graphs” are “graphs” and not ”tables”, so, please, correct appropriately the figure 1S (supplementary file). In my opinion, although both, tables and graphs, are highly informative for these data, graphs must be chosen and added to the main text.  
Comment:

Lines 140-141. See the previous comment about line 65.
Response:

In response to your comment, Figure 1S was converted to graphs and added the main text as Figure 1. Please check the revised manuscript. In addition, the following title and caption was described.

(Line 106-110)

Figure 1. Scattergrams of the MICs and ZDs of the tested cephalosporin drugs for 130 S. aureus isolates from bovine mastitis. (a) Cefazolin MICs versus CEZ30 ZDs; (b) Cefapirin MICs versus CEP10 ZDs; (c) Cefapirin MICs versus CEP30 ZDs; (d) Cefalonium MICs versus CNM10 ZDs; (e) Cefalonium MICs versus CNM30 ZDs. The dotted lines mean the ECVs established in this study. Each marker may include several strains.

Comment:

Lines 18-19 and  67. It seems obvious that 30 micrograms disk must produce higher inhibition zones than those of only 10 micrograms for the same antibiotic. Please, delete.
Comment:

Lines 145-147. See the previous comment about lines 18-19 and 67.
Response:

In response to your comment, the sentences were deleted.

Comment:

Line 83, table 3. The title of the table is confusing. The correlations studied were between disk content and MCI for each antimicrobial. Besides, I do not understand the role of the data on the last column. Could you explain the interest of the correlations between cefazolin with both cefapirin and cefalonium?
Response:

In this study, we tried to compare the results of cefapirin and cefalonium with that of cefazolin by regarding cefazolin as the standard cephalosporin drugs. However, as you indicated, the correlation between drugs is unlikely to be important. We finally agree with your comment, and then decided to remove these data from the text. In addition, Table 3 was deleted because of little informative. Please check the revised manuscript.

Comment:

Lines 22 and  88-90. I think that the closer values of the correlation coefficients for both couples of disks (10 and 30 micrograms) do not support these recommendations. What statistical test was used?
Comment:

Lines 151-154. See the previous comment about lines 22 and 88-90.

Response:

In this study, correlation coefficients were obtained by Pearson’s correlation analysis. As you indicated, clear differences may be not found in correlation coefficients between the two contents for each drug. We finally agree with you, and the corresponding sentences were removed from the text. In addition, the following sentence was added.

(Line 24-25)

We believe that both 10 μg and 30 μg CEP and CNM susceptibility disks will be helpful for guiding the appropriate use of these antibiotics for bovine mastitis.

(Line 75-77)

As the results, the zone diameters of both 10 µg and 30 µg cefapirin and cefalonium disks have high correlation with their MICs, as well as that of cefazolin disk.

(Line 155-156)

The correlation analysis indicated that both 10 μg and 30 μg cefapirin and cefalonium disks were effective for estimating drug susceptibility.

Comment:

Line 92. Why is important this non significative difference?
Response:

In response to the above comments, the sentence was deleted.

Comment:

Lines 95-96. Maybe this idea about cefazolin and detection of methicillin resistance should be added to the introduction for clarifying the use of cefazolin on this study.
Lines 97-99. It is not clear if testing cefazolin for detection of methicillin resistance in one of the objectives of this study and this should be clarified.
Line 99. It is not possible from data on table 2 verify this statement since the data on the table are not divided for methicillin resistant and susceptible isolates.
Lines 102-103. Do we have to suppose that the authors are also checking cefapirin and cefalonium for detecting methicillin resistance?
Response:

Thank you very much for your critical comment. In this study, detection of methicillin resistance by cefazolin was NOT the objective of this study because the previous studies clarified the limited detection performance by this drug. In addition, as you indicated, it would be easily estimated that detection of methicillin resistance by cefapirin and cefalonium is also incompetent. We would have liked to simply confirm that the detection performance of cefapirin and cefalonium is comparable to that of cefazolin, and thus this consideration is not necessarily essential in this study. In response to your comment, we finally decided to remove the relevant sentences from the text.

We greatly appreciate your help concerning improvement to this paper. We hope that the revised manuscript is now acceptable for publication.

Yours very sincerely,

Kazuki Harada, DVM, Ph. D.
